# Magic Clothing: Controllable Garment-Driven Image Synthesis

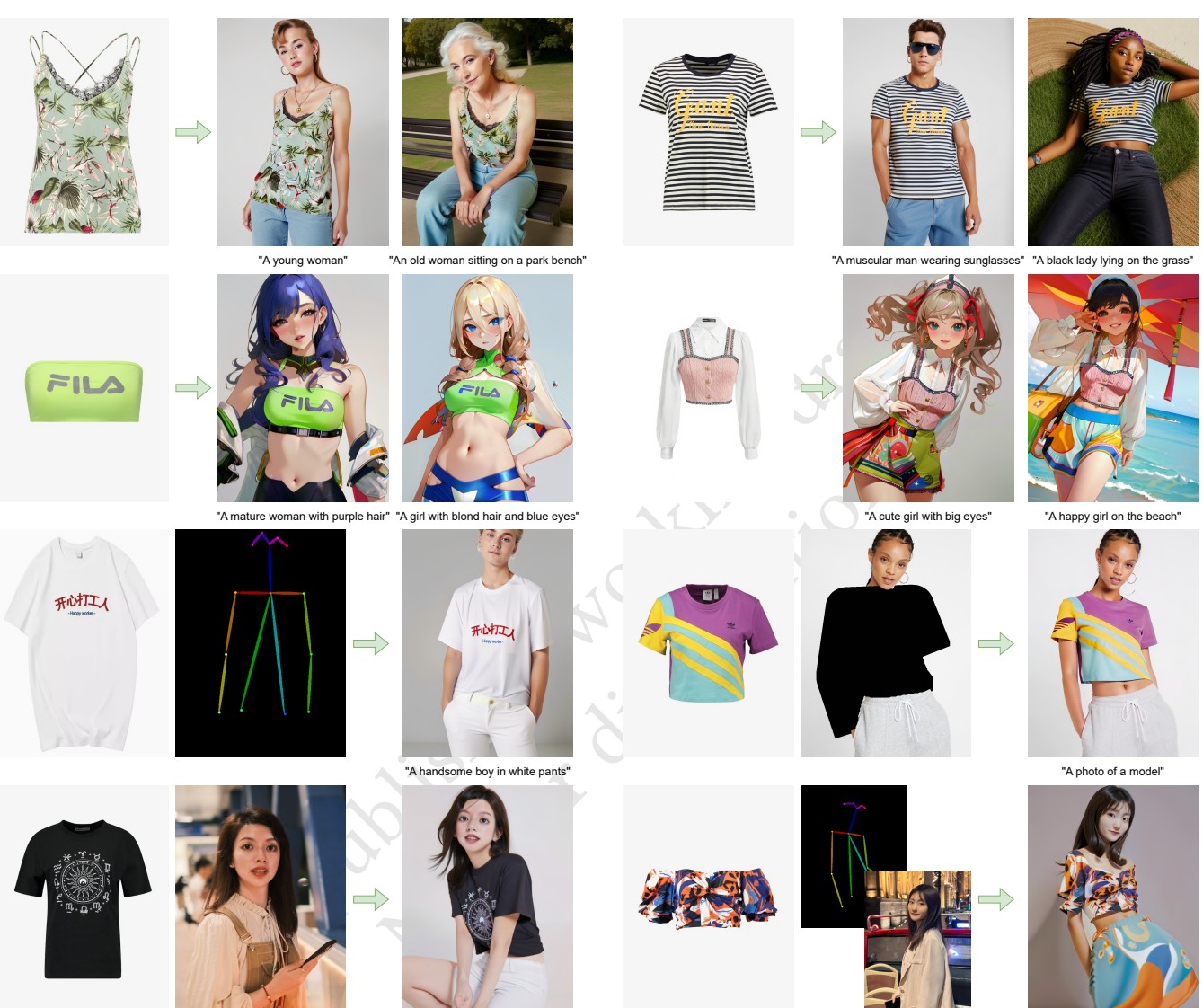

**Figure 1: Examples of our garment-driven image synthesis results. Given a target garment and a text prompt, our Magic Clothing is able to generate photorealistic (1st row) and anime-style (2nd row) characters with different finetuned latent diffusion models (LDMs) [33]. Besides, more of conditional controls can be combined with our Magic Clothing such as ControlNet [51] and IP-Adapter [50] (3rd and 4th rows).**

## ABSTRACT

We propose Magic Clothing, a latent diffusion model (LDM)-based network architecture for an unexplored garment-driven image synthesis task. Aiming at generating customized characters wearing the target garments with diverse text prompts, the image controllability is the most critical issue, i.e., to preserve the garment details and maintain faithfulness to the text prompts. To this end, we introduce a garment extractor to capture the detailed garment features, and employ self-attention fusion to incorporate them into the pretrained

LDMs, ensuring that the garment details remain unchanged on the target character. Then, we leverage the joint classifier-free guidance to balance the control of garment features and text prompts over the generated results. Meanwhile, the proposed garment extractor is a plug-in module applicable to various finetuned LDMs, and it can be combined with other extensions like ControlNet and IP-Adapter to enhance the diversity and controllability of the generated characters. Furthermore, we design Matched-Points-LPIPS (MP-LPIPS), a robust metric for evaluating the consistency of the target image to the source garment. Extensive experiments demonstrate that our Magic Clothing achieves state-of-the-art results under various conditional controls for garment-driven image synthesis. Our source code is publicly available (for the review process, please refer to our supplementary material).

## CCS CONCEPTS

• **Computing methodologies** → **Computer vision tasks**; **Computer vision problems**; *Supervised learning*.

## KEYWORDS

Garment-driven image synthesis, Latent diffusion models

**ACM Reference Format:**
. 2024. Magic Clothing: Controllable Garment-Driven Image Synthesis. In . ACM, New York, NY, USA, 10 pages. https://doi.org/10.1145/nnnnnnn.nnnnnnn

## 1 INTRODUCTION

Latent diffusion model (LDM)-based generative approaches [7, 9, 13, 15, 30, 33, 41] have achieved great success in text-to-image synthesis in recent years. Besides the textual condition, many other forms of conditional control have been explored for LDM-based image synthesis, such as pose, sketch and facial conditions [29, 43, 51]. Among these researches, little attention has been paid to image synthesis conditioned on a specific garment so far, which is a promising task with enormous application prospects for e-commerce and metaverse, etc. Such a garment-driven image synthesis task aims to generate a character wearing the target garment according to the customized text prompt, where the image controllability is the most critical issue. More specifically, the main challenges of garment-driven image synthesis include preserving the garment details and maintaining faithfulness to the text prompts.

Previous approaches to traditional subject-driven image synthesis [12, 19, 20, 25, 35, 38, 42, 54] usually focus on overall conditional information like appearance and structure, and invert the subject image into the text-embedding space, which are insufficient to capture the complicated fine-grained features of the garments. Another popular task similar to garment-driven image synthesis is virtual try-on (VTON) [8, 14, 21, 27, 28, 45, 48], aiming to generate a specific person wearing the target clothes. However, VTON is more of an image-inpainting task, which only requires faithfully preserving the target garment features without any creative capability of following arbitrary customized text prompts.

In view of the aforementioned challenges, we introduce Magic Clothing, an LDM-based network architecture that focuses on character generation conditioned on the given garments and text prompts. To preserve the fine-grained garment details, we propose

a garment extractor with the UNet architecture [34]. By leveraging the power of pretrained LDMs [33], it allows smooth incorporation of detailed garment features into the denoising UNet through self-attention fusion. To maintain faithfulness to arbitrary customized text prompts, we randomly drop garment features and text prompts from a joint distribution in training to enable the joint classifier-free guidance, which effectively balances the control of garment features and text prompts over the generated results. Our garment extractor is also a plug-in module compatible with various finetuned LDMs or extensions like ControlNet [51] and IP-Adapter [50] to further control the pose, face and even style of the character without degrading the garment details. In practice, we propose a novel robust metric, namely Matched-Points-LPIPS (MP-LPIPS) to quantify the garment-driven image synthesis quality. It measures the consistency of the target image to the source garment by comparing the patches obtained from point matching, thus mitigating the undesired effect of pose and background on the evaluation.

In summary, the main contributions of this work are as follows:

- Magic Clothing is, to the best of our knowledge, the first LDM-based work to investigate the unexplored garment-driven image synthesis task.
- We propose a garment extractor to incorporate garment features into the denoising process via self-attention fusion. And the joint classifier-free guidance is applied to balance the control of garment features and text prompts.
- Our garment extractor is a plug-in module applicable to various finetuned LDMs, which can be easily combined with other powerful extensions, such as ControlNet [51] or IP-Adapter [50], to employ additional conditions.
- We develop a robust metric namely MP-LPIPS to evaluate the consistency of the target image to the source garment. Qualitative and quantitative experiments demonstrate our state-of-the-art performance on garment-driven image synthesis with high controllability.

## 2 RELATED WORK

### 2.1 Latent Diffusion Models

Latent diffusion models (LDMs) [33] have been successfully used for text-to-image generation tasks. Based on this robust foundational model, various LDM-based researches and applications have emerged. To add spatial conditioning controls to the pretrained LDMs, ControlNet [51] incorporated trainable encoder blocks into the original UNet. Meanwhile, T2I-Adapter [29] proposed a compact network design that provided the same functionality as ControlNet but with reduced complexity. To further reduce the training cost, Uni-ControlNet [55] proposed a unified framework that handles different conditional controls in a flexible and composable manner within one single model. On the other hand, LDMs have also played a significant role in the domain of image editing. InstructPix2Pix [3] retrained the UNet of LDMs by adding extra input channels to the first convolutional layer on a large dataset of image editing examples to make it follow the edit instructions. MasaCtrl [4] converted the self-attention in diffusion models into mutual self-attention to enable consistent image generation and complex non-rigid image editing simultaneously without additional training cost. InfEdit [46] performed consistent and faithful image editing for both rigid and

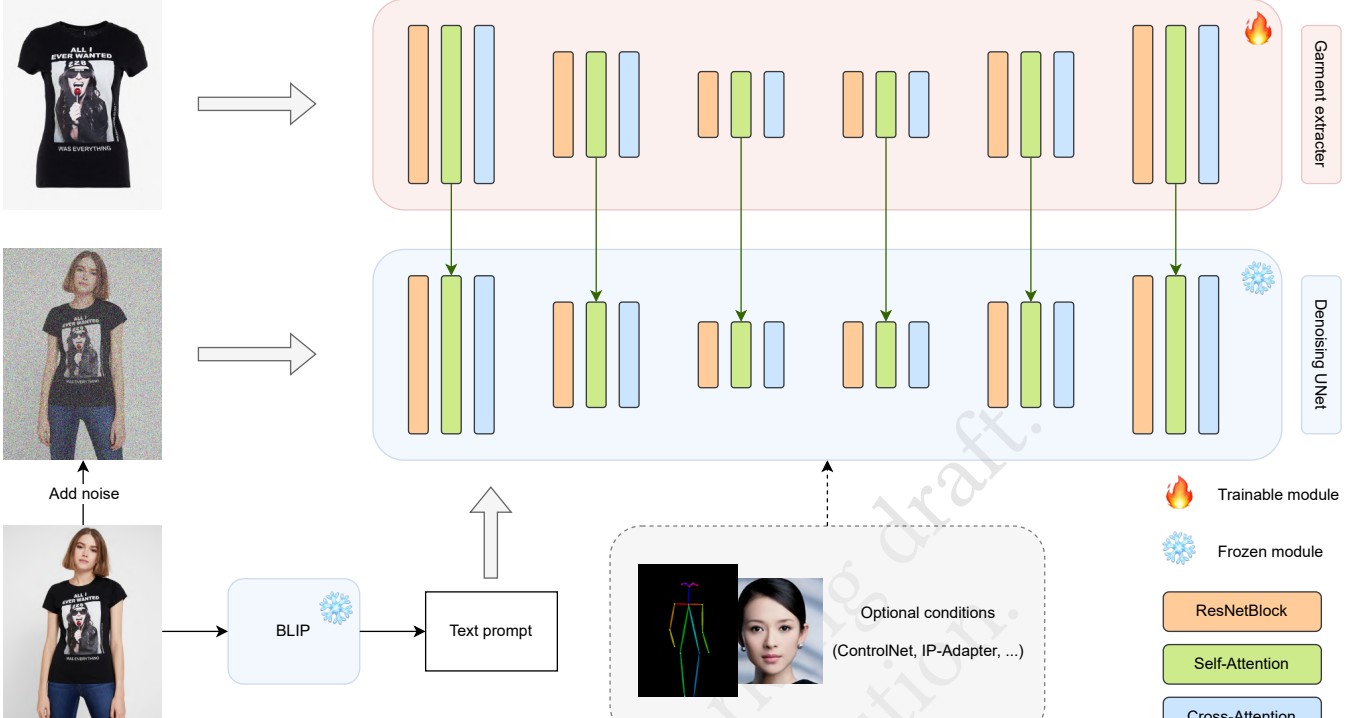

**Figure 2: Overview of our Magic Clothing. We propose a garment extractor that captures the garment features and incorporate these features into the denoising process in self-attention layers. Besides the paired garment and character images, we obtain the text prompts for training through BLIP [23]. Only the garment extractor requires additional training, which is a plug-in module compatible with other useful extensions like ControlNet [51] or IP-Adapter [50].**

non-rigid semantic changes through the denoising diffusion consistent model and attention control mechanisms. In this paper, we make full use of the power of pretrained LDMs in text-to-image generation, and introduce an extra garment extractor for garment-driven image synthesis.

## 2.2 Subject-Driven Image Synthesis

Given an image of a particular subject, the subject-driven image synthesis task is to generate a novel image in different contexts while maintaining high fidelity to its key visual features. Textual Inversion [12] learned to represent the concept through new "words" in the embedding space of a frozen text-to-image model, which can be composed into natural language sentences and generate customized images. DreamBooth [35] finetuned the pretrained LDMs model with the newly designed class-specific prior preservation loss to make it learn to bind a unique identifier with that specific subject. However, finetuning the entire UNet for each subject raises significant computational cost. To address this, HyperDreamBooth [36] proposed a hypernetwork capable of efficiently generating an initial prediction of a subset of network weights and significantly accelerated the finetuning process. BLIP-Diffusion [22] introduced a new multimodal encoder which is pretrained to provide subject representation. By leveraging such visual representation, diffusion models can yield a wide range of subject variations. IP-Adapter [50]

designed a decoupled cross-attention mechanism that separates cross-attention layers for text features and image features. Despite its simplicity, this method achieved comparable or even better performance than specifically finetuned models. Versatile Diffusion [47] expanded the existing single-flow diffusion pipeline into a multitask multimodal network to enable cross-modal generality. Break-A-Scene [2] extracted a distinct text token for each concept from a single image by introducing a two-phase customization process that optimizes a set of dedicated textual embedding and the model weights. In this work, we focus on an unexplored garment-driven image synthesis task, which requires much better detail preservation together with the creative capability following diverse text prompts.

## 3 METHOD

### 3.1 Preliminary

Unlike other pixel-space based diffusion models, latent diffusion models (LDMs) [33] are designed to perform the denoising process in the latent space for reducing the computational cost. Our approach is based on the LDM framework, which consists of three main networks, i.e. Vector Quantized Variational AutoEncoder (VQ-VAE) [11], CLIP VIT-L/14 [32], and a denosing UNet [34]. The VQ-VAE enables image representation in the latent space, whose encoder $\mathcal{E}$ compresses the original image $\mathbf{x}$ into the image latent $\mathbf{z}$

and decoder $\mathcal{D}$ reconstructs the image $\mathbf{x}$ from the image latent $\mathbf{z}$ with minor information loss. The CLIP VIT-L/14 $\mathcal{T}$ is utilized to convert the original text prompts $\mathbf{y}$ into token embeddings $\mathcal{T}_{\mathbf{y}}$. During training, the noise $\epsilon$ corresponding to the time step $t$ is added to the image latent $\mathbf{z}$ to get a noise latent $\mathbf{z}_t$, and the denoising UNet $\epsilon_\theta$ is designed to predict the input noise $\epsilon$ given the noise latent $\mathbf{z}_t$, time step $t$ and the token embeddings $\mathcal{T}_{\mathbf{y}}$. The optimization process is performed by minimizing the following loss function:

$$\mathcal{L} = \mathbb{E}_{\mathcal{E}(\mathbf{x}), \mathcal{T}_{\mathbf{y}}, \epsilon \sim \mathcal{N}(0,1), t} \left[ \| \epsilon - \epsilon_\theta(\mathbf{z}_t, t, \mathcal{T}_{\mathbf{y}}) \|_2^2 \right]. \tag{1}$$

## 3.2 Network Architecture

Figure 2 presents an overview of our method. During training, we convert the character image $\mathbf{I}_C \in \mathbb{R}^{3 \times H \times W}$ and the garment image $\mathbf{I}_G \in \mathbb{R}^{3 \times H \times W}$ into the image latents $\mathbf{z}_C, \mathbf{z}_G \in \mathbb{R}^{4 \times \frac{H}{8} \times \frac{W}{8}}$, respectively, using VAE Encoder $\mathcal{E}$. On the other hand, we caption $\mathbf{I}_C$ with BLIP [23] to obtain the text prompt $\mathbf{y}$ and transform it into token embeddings $\mathcal{T}_{\mathbf{y}}$. Transferring knowledge from the image synthesis to feature extraction, we introduce a garment extractor $\mathcal{E}_G$ that has the same UNet architecture as the denoising UNet to extract the detailed garment features. Inspired by the powerful spatial-attention mechanisms [6, 18, 48, 49], we incorporate the extracted garment features into the original denoising process smoothly through self-attention fusion. More concretely, $\alpha_i$ and $\beta_i$ refer to the normalized attention hidden states of the $i$-th self-attention block in the original denoising UNet $\epsilon_\theta$ and the garment extractor $\mathcal{E}_G$, respectively. The calculation of self-attention in $\epsilon_\theta$ after incorporation of garment features is modified as:

$$\text{Attention}(\alpha_i, \beta_i) = \text{Softmax}\left( \frac{\mathbf{W}_Q \beta_i (\mathbf{W}_K [\alpha_i, \beta_i])^T}{\sqrt{d}} \right) \mathbf{W}_V [\alpha_i, \beta_i], \tag{2}$$

where $[\cdot]$ denotes concatenation operation, $d$ is the feature dimension, and $\mathbf{W}_Q, \mathbf{W}_K, \mathbf{W}_V$ are linear projection weights of query, key, and value in self-attention layers.

To maintain the text-to-image synthesis capabilities of the original LDM and reduce the training cost, we keep the weights of $\epsilon_\theta$ frozen. Then we only train our garment extractor with its weights initialized by the weights of $\epsilon_\theta$, which further speeds up the training process. Our training objective is summarized as:

$$\mathcal{L} = \mathbb{E}_{\mathbf{z}_C, \beta, \mathcal{T}_{\mathbf{y}}, \epsilon \sim \mathcal{N}(0,1), t} \left[ \| \epsilon - \epsilon_\theta(\mathbf{z}_{C_t}, \beta, t, \mathcal{T}_{\mathbf{y}}) \|_2^2 \right], \tag{3}$$

where $\mathbf{z}_{C_t}$ is obtained by adding noise to the character image latent $\mathbf{z}_C$ at time step $t$, and $\beta$ is the overall garment features from self-attention blocks of our garment extractor $\mathcal{E}_G$.

During inference, given an input garment image and a text prompt for the character, Magic Clothing is capable of generating an image of the character wearing the target garment. To add more conditional controls, our garment extractor can be utilized with other LDMs and extensions like ControlNet and IP-Adapter, which will be discussed in Section 3.4. Meanwhile, the garment features are shared across all denoising steps, adding minimal computational cost to the original LDM inference process.

## 3.3 Joint Classifier-free Guidance

Classifier-free guidance [16] is a method that helps the diffusion models to attain a trade-off between sample quality and diversity

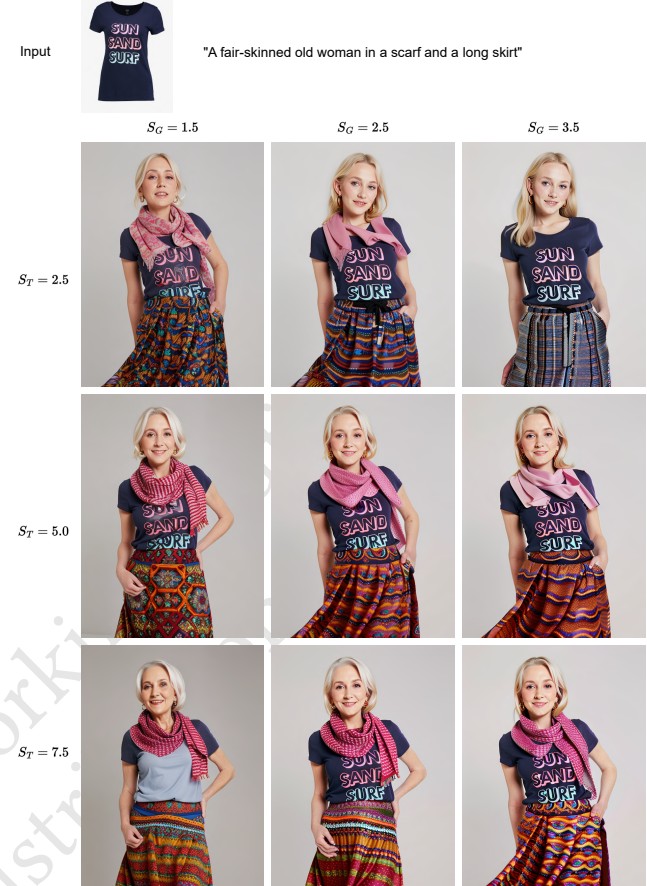

**Figure 3: Example results with different text guidance scales $S_T$ and garment guidance scales $S_G$. With a larger $S_T$, the generated image becomes more faithful to the text prompt. While with a larger $S_G$, more garment details are preserved.**

by jointly training a conditional and an unconditional diffusion model. The implementation during training is relatively simple by setting the conditional control $\mathbf{c} = \varnothing$ with some probability. During inference, the score estimate $\hat{\epsilon}_\theta(\mathbf{z}_t, \mathbf{c})$ linearly combine the conditional and unconditional predictions as the following equation:

$$\hat{\epsilon}_\theta(\mathbf{z}_t, \mathbf{c}) = \epsilon_\theta(\mathbf{z}_t, \varnothing) + s \cdot (\epsilon_\theta(\mathbf{z}_t, \mathbf{c}) - \epsilon_\theta(\mathbf{z}_t, \varnothing)), \tag{4}$$

where $s \geq 1$ represents the strength of conditional controls.

As for our garment-driven image synthesis task, two variant conditional controls should be considered: detailed garment features $c_G$ and text prompts $c_T$. If we consider them as independent controls, we can naively modify the independent classifier-free guidance in [24] for our use. Specifically, garment features $\mathbf{c}_G$ and text prompts $\mathbf{c}_T$ can be independently set to $\varnothing$ with some probability during training. Then at inference time, we introduce the garment guidance scale $S_G$ and text guidance scale $S_T$ to adjust the strengths of conditional controls from the garment and text prompts, respectively. The score estimate with independent classifier-free guidance

is calculated as:

$$\hat{\epsilon}_\theta(\mathbf{z}_t, \mathbf{c}_G, \mathbf{c}_T) = \epsilon_\theta(\mathbf{z}_t, \varnothing, \varnothing)$$
$$+ S_G \cdot (\epsilon_\theta(\mathbf{z}_t, \varnothing, \mathbf{c}_T) - \epsilon_\theta(\mathbf{z}_t, \varnothing, \varnothing)) \quad (5)$$
$$+ S_T \cdot (\epsilon_\theta(\mathbf{z}_t, \mathbf{c}_G, \varnothing) - \epsilon_\theta(\mathbf{z}_t, \varnothing, \varnothing)).$$

Despite its simplicity, fusing two denoising scores as done in Equation 5 could lead to undesired results since this method ignores the fact that our two conditional controls may have overlapping semantics.

Drawing inspiration from [3], we leverage joint classifier-free guidance to balance these two conditional controls. Here joint indicates that we set the drop rate of garment features and text prompts according to a joint distribution. More precisely, we randomly set 5% of training samples with $\mathbf{c}_G = \varnothing_G$, 5% with $\mathbf{c}_T = \varnothing_T$ and another 5% with both $\mathbf{c}_G = \varnothing_G$ and $\mathbf{c}_T = \varnothing_T$. Our joint classifier-free guidance score estimate is:

$$\hat{\epsilon}_\theta(\mathbf{z}_t, \mathbf{c}_G, \mathbf{c}_T) = \epsilon_\theta(\mathbf{z}_t, \varnothing, \varnothing)$$
$$+ S_G \cdot (\epsilon_\theta(\mathbf{z}_t, \mathbf{c}_G, \varnothing) - \epsilon_\theta(\mathbf{z}_t, \varnothing, \varnothing)) \quad (6)$$
$$+ S_T \cdot (\epsilon_\theta(\mathbf{z}_t, \mathbf{c}_G, \mathbf{c}_T) - \epsilon_\theta(\mathbf{z}_t, \mathbf{c}_G, \varnothing)).$$

In Figure 3, we show the effects of these two parameters on generated samples with the random seed fixed. It is noticeable that the garment in the character become more similar to the input garment with a larger $S_G$ and a larger $S_T$ will make the generated result follow the text prompt more precisely. Since a large gap between $S_T$ and $S_G$ may distort the garment details, we empirically set $S_T = 7.5$ and $S_G = 2.5$ in our experiments.

### 3.4 Plug-in Mode

Since we keep the weights of the pretrained LDM [33] frozen and only train our garment extractor $\mathcal{E}_G$, we can treat $\mathcal{E}_G$ as a plug-in module and combine it with various finedtuned LDMs to enhance the diversity of generated characters. For instance, when combined with various LoRA [17] or full-parameter finetuned LDMs, we are able to create characters in different styles, such as science fiction, realistic, and anime styles, etc. Moreover, our controllable image synthesis process is compatible with other advanced LDM extensions. To generate characters wearing the given garments with target poses, we can combine ControlNet-Openpose [51] with our Magic Clothing. And with ControlNet-Inpaint [51], our model is able to perform the virtual try-on (VTON) task [8, 21, 27, 45, 48] and generate high-fidelity results. For the purpose of generating a specific person wearing the target garment, we can combine Magic Clothing with IP-Adapter-FaceID [50], which takes the given portrait as an input condition. We remark that the finetuned LDMs and multiple extensions like ControlNet [51] and IP-Adapter [50] can be simultaneously combined with our Magic Clothing. In this way, we achieve comprehensive conditional controls over the image synthesis process, including text prompts, garments, styles, faces and postures, etc. The superior performance of our plug-in models on controllable image synthesis under various conditions is shown in Section 4.3.3.

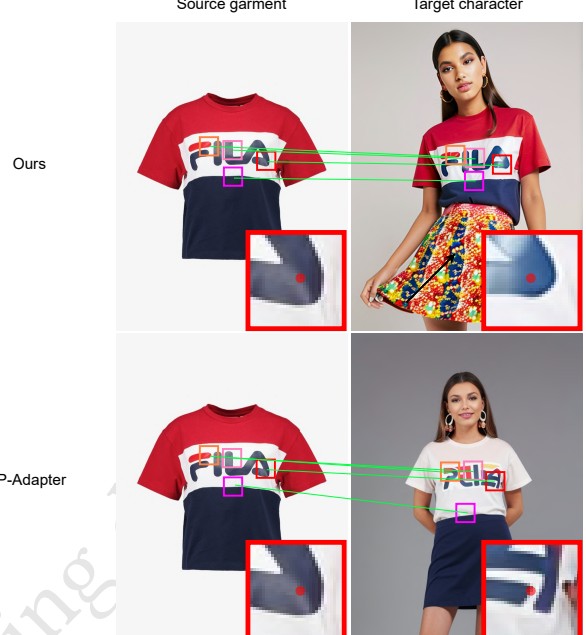

Source garment     Target character

**Figure 4: MP-LPIPS measures the consistency of the character (right column) to the garment (left column) by comparing patches centred on matched points. Given points in the source garment, we use diffusion features [40] to retrieve corresponding points in the target character.**

## 4 EXPERIMENTS

### 4.1 Experimental Setup

*4.1.1 Dataset.* To train our Magic Clothing for this new task of garment-driven image synthesis, we need not only the paired garment and character images, but also text prompts containing the character descriptions. Since the VITON-HD dataset [8] only contains the paired data of frontal half-body characters and corresponding upper-body garments, we obtain the text prompts by captioning these characters using BLIP [23].

To evaluate the model performance, we provide 5 test text prompts covering different ages, appearances, ornaments and backgrounds of the target characters using GPT-4 [1]. And we select the garments in the test set of the VITON-HD dataset as our test garments, which contains 2,032 upper-body garments. We generate 5 character images for each test garment with the provided 5 text prompts, resulting in a total of 10,160 images for evaluation.

*4.1.2 Compared Methods.* We compare our results with three state-of-the-art LDM-based subject-driven image synthesis models, IP-Adapter [50], BLIP-Diffusion [22], and Versatile Diffusion [47]. In addition, we train a ControlNet [51] that takes the garment directly as an input condition and generates images according to text prompts, namely ControlNet-Garment. For fair comparison, all the experiments are conducted at the resolution of $768 \times 576$.

*4.1.3 Evaluation Metrics.* To evaluate the garment fidelity, the details of the garment on the generated character must be compared

with those of the original garment. Following previous subject-driven methods [35, 36], we use CLIP-I [32] to compute the similarity of the same subject in different images. However, this metric is sensitive to background and the character pose in our task. Recently, S-LPIPS [53] is designed to tackle the problem of changes in the shape and posture of generated characters, but it still relies on human pose estimation [5, 39, 44] and paired images. Surprisingly, it is reported in [40] that geometric corresponding points can be accurately predicted from the intermediate layer activations of pretrained LDMs, even though they are not trained with any explicit geometry supervision. Driven by the above prospects and challenges, we propose Matched-Points-LPIPS (MP-LPIPS) that measures the LPIPS distance [52] between pixels on garment $\mathbf{I}_G$ and matched pixels on generated character $\mathbf{I}_C$. Specifically, we uniformly select N points $\mathbb{P}_G$ on each garment, calculating their diffusion features [40] $\mathcal{F}_{\mathbf{t,l}}(\mathbb{P}_G)$ at time step $t$ and layer $l$. Then we find the matched points $\mathbb{P}_C$ on the target character by calculating cosine similarity as:

$$\mathbb{P}_C = \arg\min_{\mathbb{P}} \frac{\mathcal{F}_{\mathbf{t,l}}(\mathbb{P}_G)\mathcal{F}_{\mathbf{t,l}}(\mathbb{P})}{\|\mathcal{F}_{\mathbf{t,l}}(\mathbb{P}_G)\|\|\mathcal{F}_{\mathbf{t,l}}(\mathbb{P})\|}, \tag{7}$$

and finally define MP-LPIPS distance as:

$$d_{\text{MP-LPIPS}} = \frac{1}{N} \sum_{i=1}^{N} d_{\text{LPIPS}}\left(\mathbf{P}_G^i, \mathbf{P}_C^i\right). \tag{8}$$

We measure the average LPIPS distance of the patches $\mathbf{P}_G, \mathbf{P}_C \in \mathbb{R}^{N \times H \times W}$ centred on N pairs of points $\mathbb{P}_G$ and $\mathbb{P}_C$, where $H$ and $W$ indicate the height and width of the patch, respectively. As shown in Figure 4, MP-LPIPS effectively measures the consistency of the garment to the character without the need for manual annotation. More importantly, it is robust to factors such as the background and posture. More detailed settings are to be provided in the supplementary material.

In addition, we use CLIP-T [32] to measure the faithfulness of results to text prompts and CLIP aesthetic score [37] to evaluate the general image quality.

## 4.2 Implementation Details

In our experiments, we initialize the weights of our garment extractor by inheriting the pretrained weights of the UNet in Stable Diffusion v1.5 [33], and only finetune its weight while keeping the weights of other modules frozen. Our model is trained on the paired images from VITON-HD [8] dataset at the resolution of 768 × 576 and the corresponding captions that obtained from BLIP [23]. We adopt the AdamW optimizer [26] with a fixed learning rate of 5e-5. The model is trained for 100,000 steps on a single NVIDIA A100 GPU with a batch size of 16. At inference time, the images are generated with the UniPC sampler [56] for 20 sampling steps.

## 4.3 Experimental Results

*4.3.1 Qualitative Results.* Figure 5 presents qualitative results with our text prompts and garment images from the VITON-HD dataset [8]. We can see that IP-Adapter [50] closely follows the textual conditions and preserves the general appearance of the garment. However, it struggles to preserve the garment details such as printed patterns or texts. BLIP Diffusion [22] and Versatile Diffusion [47] are

**Table 1: Quantitative comparison with traditional subject-driven image synthesis methods. The best and second best results are reported in bold and underline, respectively.**

| Method | MP-LPIPS ↓ | CLIP-T ↑ | CLIP-I ↑ | CLIP-AS ↑ |
|---|---|---|---|---|
| ControlNet-Garment | 0.414 | 0.323 | 0.636 | 5.511 |
| Versatile Diffusion | 0.277 | 0.240 | 0.790 | 5.242 |
| BLIP-Diffusion | 0.224 | 0.233 | 0.765 | 5.316 |
| IP-Adapter | 0.194 | 0.289 | 0.760 | 5.426 |
| **Magic Clothing** | **0.143** | **0.336** | **0.803** | **5.526** |

**Table 2: Ablation study of different classifier-free guidance (CFG). The best results are reported in bold.**

| Method | MP-LPIPS ↓ | CLIP-T ↑ | CLIP-I ↑ | CLIP-AS ↑ |
|---|---|---|---|---|
| Independent CFG | 0.177 | 0.300 | 0.772 | 5.353 |
| **Joint CFG** | **0.143** | **0.336** | **0.803** | **5.526** |

weak at following text prompts and also have difficulty maintaining garment details. To make matters worse, they tend to generate garment variations instead of characters when the garment details are complex, as shown in the last line of the Figure 5. These traditional subject-driven image synthesis methods typically embed the image in the CLIP text embedding space, which emphasis on overall structural similarity rather than preservation of fine-grained details. However, humans are sensitive to subtle variations in garment features such as printed patterns or text, making these conventional methods inadequate for the garment-driven image synthesis task. Although the results of ControlNet [51] follow the text prompts reasonably well, they almost ignore the conditional control from the garments, resulting in the garments on the generated characters being completely different from the input garments. We believe that this is mainly due to ControlNet failing to learn the spatial correlations between the garments and the characters during its training process. Compared to other methods, the results from our Magic Clothing not only maintain faithfulness to the text prompts but also preserve details of the given garment, demonstrating the best performance in garment-driven image synthesis.

*4.3.2 Quantitative Results.* Table 1 shows the quantitative results with our text prompts and garment images from the VITON-HD dataset [8]. ControlNet [51] achieves high scores on CLIP-T and CLIP-AS, which demonstrates its ability to follow text prompts and generate images with high aesthetic quality. However, it fails to retain garment details due to the spatial mismatch between the given garment and target character. According to their CLIP-I scores, Versatile Diffusion [47], Blip Diffusion [22] and IP-Adapter [50] can generate characters that are more similar to the person wearing the garment in the VITON-HD dataset. Nonetheless, they extract image features using the CLIP image encoder [32], which captures only semantic information and loses garment details, resulting in inferior performance on MP-LPIPS. In comparison, our Magic Clothing generates characters with fine-grained garment details and text prompt fidelity, significantly outperforming other methods on all the evaluation metrics.

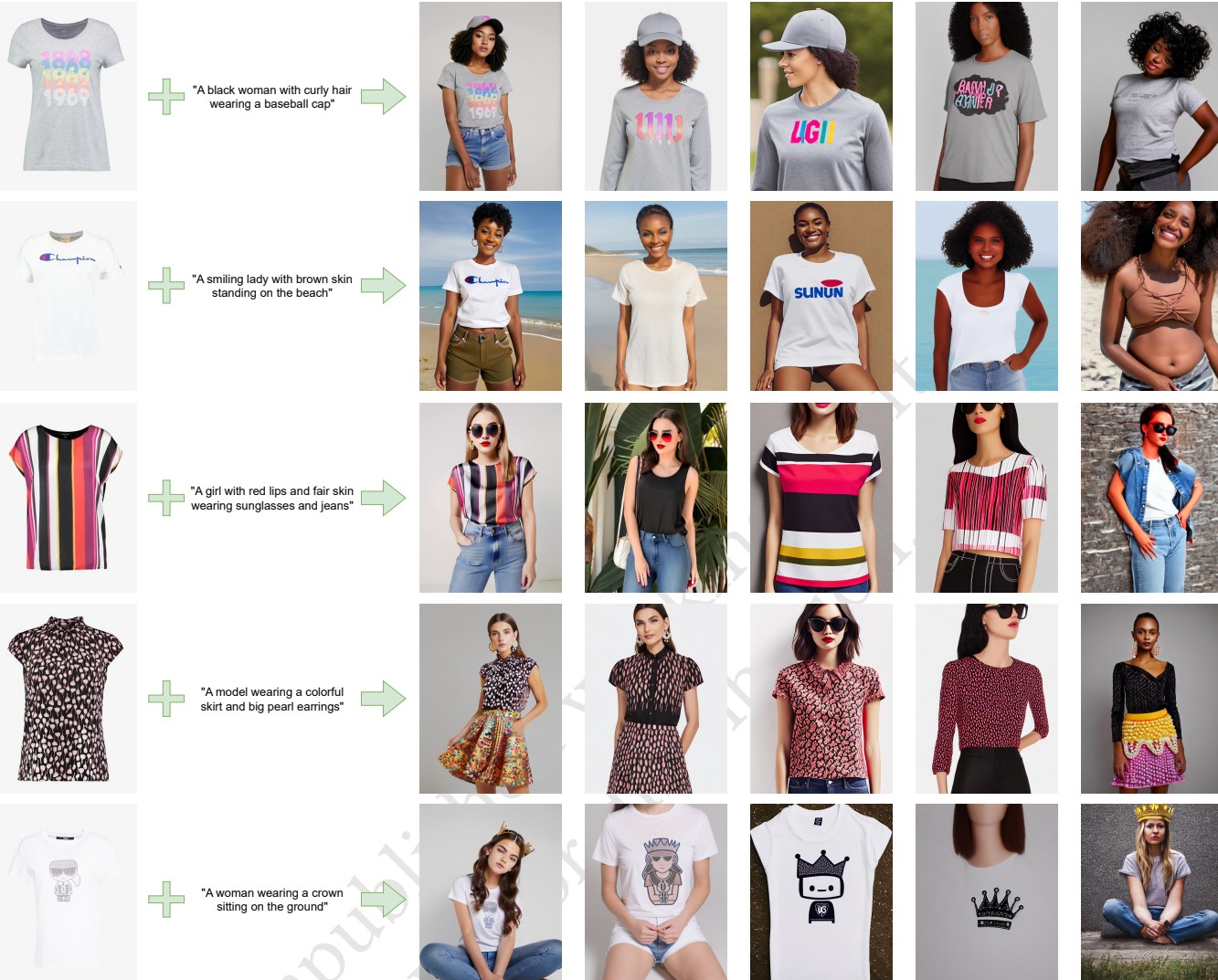

**Figure 5: Qualitative comparison with traditional subject-driven image synthesis methods, including IP-Adapter [50], BLIP-Diffusion [22], Versatile Diffusion [47], and ControlNet-Garment [51].**

*4.3.3 Plug-in Results.* Figure 6 presents the results of our garment extractor as a plug-in module combined with other finetuned LDMs and extensions. When built upon LDMs [33] finetuned for animate-style image synthesis, Magic Clothing is able to generate anime characters wearing target garments (1st row). Further condition controls can be added by combining it with other advanced extensions. For example, we can arbitrarily change the poses of the characters (2nd row) with the help of ControlNet-Openpose [51]. While with ControlNet-Inpaint [51] (3rd row), Magic Clothing can accomplish the traditional virtual try-on task. With the help of IP-Adapter-FaceID [50], we can specify the identity of the characters wearing the target garments (4th row). More remarkably, multiple extensions can be simultaneously combined with Magic Clothing to create a wide variety of customized images (5th row).

## 4.4 Ablation Study

To verify the effectiveness of our joint classifier-free guidance, we train a model with the same network architecture but using independent classifier-free guidance as described in Section 3.3. In contrast to our method, we set the drop rate of garment features and text prompts independently to 10% during training. As summarized in Table 2, our joint classifier-free guidance is substantially better than the independent classifier-free guidance on all metrics. Benefiting from the joint setting, our LDM-based model achieves a better training balance between conditional and unconditional denoising. The independent setting, on the other hand, largely favours the conditional denoising with respect to both or one of the garment and the text prompt. Note that since the text prompt for training may naturally contain the information about garment, these two

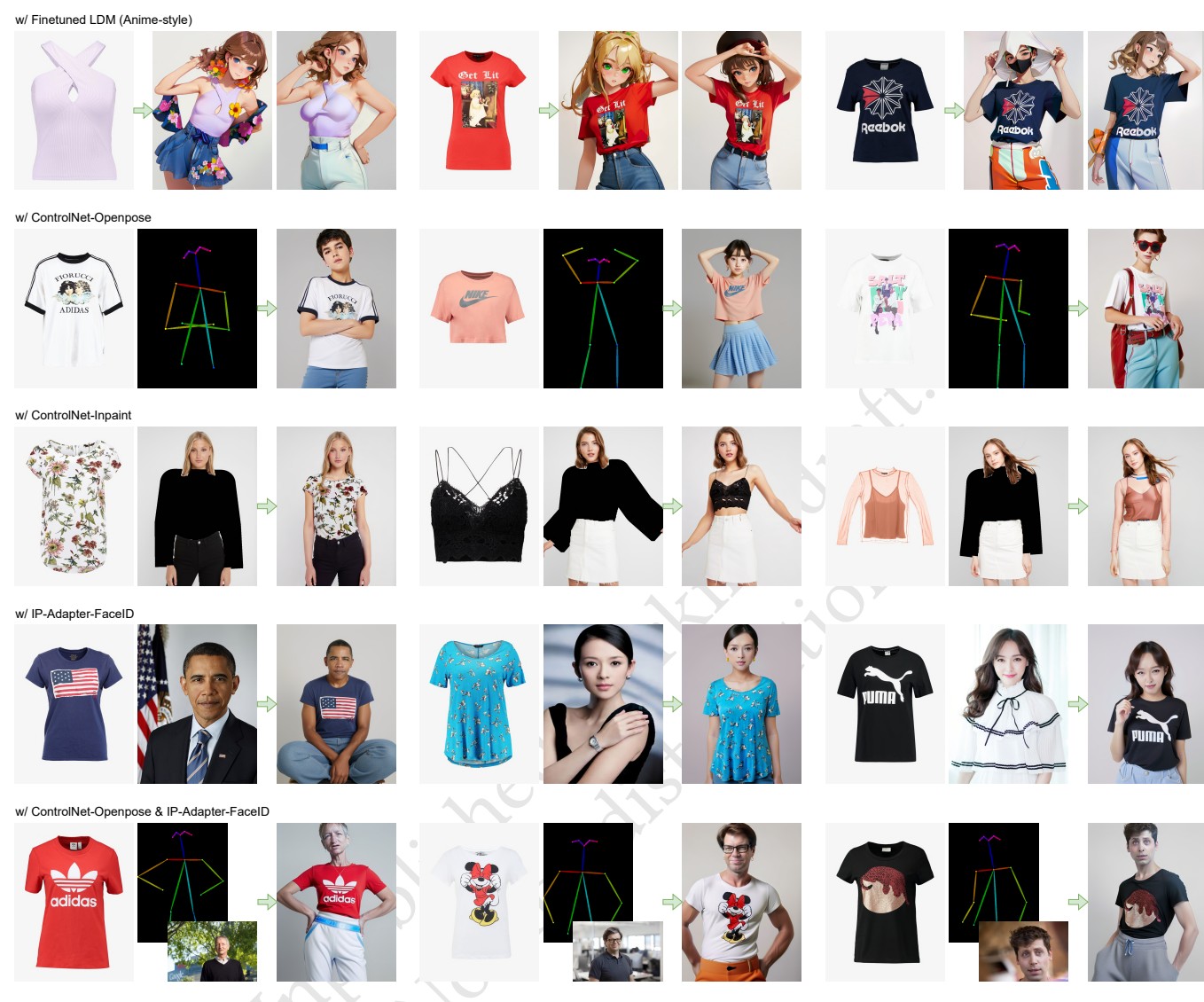

**Figure 6: Examples of plug-in results of our Magic Clothing combined with finetuned anime-style LDMs (1st row), ControlNet-Openpose (2nd row), ControlNet-Inpaint (3rd row), IP-Adapter-FaceID (4th row), and multiple extensions (5th row).**

conditional controls can hardly be independent from each other and are better considered jointly.

## 4.5 Limitation

Despite the state-of-the-art performance in garment-driven image synthesis, limitations remain in our Magic Clothing. For instance, the quality of our generated images are highly dependent on the base diffusion models. Further improvement can be achieved by using more powerful pretrained models like SDXL [31] and Stable Diffusion 3 [10]. Another limitation is that due to the limited training samples in the VITON-HD dataset [8], Magic Clothing may fail to generate perfect results for complicated garments such as down jackets and coats. A possible solution in the future is to collect more comprehensive dataset for training our garment extractor.

## 5 CONCLUSION

This paper presents Magic Clothing, an LDM-based network architecture for the unexplored garment-driven image synthesis task. A garment extractor is employed to incorporate the garment details via self-attention fusion, and the joint classifier-free guidance is applied to balance the control of garment features and text prompts. Our garment extractor is a plug-in module which can be easily combined with other useful extensions for additional conditions. Comprehensive experiments demonstrate our superiority in generating diverse images controlled by the given garments and text prompts, implying the tremendous potential of controllable garment-driven image synthesis.

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
