# OpenReview forum: "Magic Clothing: Controllable Garment-Driven Image Synthesis"
_acmmm.org/ACMMM/2024/Conference — MM2024 Poster_

### Official Review · Reviewer_Ewo2 · 2024-05-24

**Rating:** 3
**Confidence:** 4

**Summary:**

This work investigates an unexplored garment-driven image synthesis task, presenting a novel network architecture to incorporate the garment-image features into latent diffusion models while maintaining faithfulness to the text prompts.

**Strengths:**

+1. This paper proposes Magic Clothing, a latent diffusion model (LDM)-based network architecture for an unexplored garment-driven image synthesis task.

+2.  This paper proposes a garment extractor to incorporate garment features into the denoising process via self-attention fusion. The joint classifier-free guidance is applied to balance the control of garment features and text prompts.

+3. This paper develops a robust metric namely MP-LPIPS to evaluate the consistency of the target image to the source garment.

**Limitations:**

1). Although the methods proposed in this paper have shown promising results, the author has failed to clearly explain the fundamental differences between this method and previous methods, and why it produces favorable outcomes. For example, what are the specific differences between this method and OOTDiffusion (OOTDiffusion: Outfitting Fusion-based Latent Diffusion for Controllable Virtual Try-on)? What quality improvements do these differences bring about?

2). This method is pretty much an image-based virtual try-on task, so why do the authors consider this an unexplored new task, the claimed garment-driven image synthesis task?

3). It is regrettable that this method has not been compared with any image-based virtual try-on baseline methods.

4). The paper lacks appearance similarity metrics for quantitative experiments, such as SSIM.

5). The ablation experiments lack depth.

6).  There is a lack of visualized ablation experiments.

7).  The absence of coat samples in VITON-HD, as claimed by the authors: "Magic Clothing may fail to generate perfect results for complicated garments such as down jackets and coats.", is only unconvincing due to dataset limitations.

8). The author's work leans more towards addressing engineering problems, and the entire network still remains within the confines of current mature virtual try-on architectures. Furthermore, comparing a general architecture with a specialized one in experiments is unfair, so it is suggested to use specialized baselines (e.g. StableVITON) to further demonstrate the effectiveness of the proposed method.

**Suitability:**

2

---

### Official Review · Reviewer_G6f2 · 2024-05-24

**Rating:** 5
**Confidence:** 4

**Summary:**

This paper introduces "Magic Clothing", a novel latent diffusion model (LDM) based network architecture designed for garment-driven image synthesis. Key contributions include: a garment extractor that captures detailed garment features and integrates them into pretrained LDMs via self-attention fusion, ensuring detailed garment preservation and adherence to text prompts; the garment extractor is a plug-in compatible with various finetuned LDMs and extensions like ControlNet and IP-Adapter; a robust evaluation metric, MP-LPIPS, to evaluate the consistency between the generated image and the source garment. Extensive experiments show that the proposed method achieves superior performance in generating customized characters wearing target garments with high controllability and fidelity.

**Strengths:**

1. **Well-written**: This paper is crafted organized and well-written.
2. **Diverse experiments and evaluations**: This paper includes rich experimental results and evaluations that provide strong evidence of the effectiveness of the proposed method.
3. **Garment extractor**:  The garment extractor module is one of the key innovations in the proposed method, that accurately captures and preserves the complex details of the input garments.

**Limitations:**

1. **Garments mismatched for men**: Due to VITON-HD containing only frontal views of women and top clothing image pairs, the generation results (from male faces) in the last row of Figure 6 show obvious female characteristics, such as earrings, makeup, etc. It would be better to discuss this mismatch issue.

2. **Limitations on simple posture**: The proposed method outperforms the existing works on "subject-driven image generation" and "virtual try-on". However, using the generated human images instead of real human images, while increasing the garment controllable rate to a certain extent and reducing the generation error rate, also reduces the diversity of character postures.

3. **Reference dataset**: Due to the VITON-HD dataset having limitations on gender and garment generation (only upper body), I suggest authors also make comparisons between proposed methods and existing works on the DressCode dataset [1], which contains rich garments, images, and multiple categories (upper body, lower body, and dresses).

   [1] Morelli, Davide, et al. "Dress code: high-resolution multi-category virtual try-on." CVPR 2022

**Suitability:**

3

---

### Official Review · Reviewer_jEoh · 2024-05-24

**Rating:** 3
**Confidence:** 4

**Summary:**

This work introduces a novel network architecture based on the Latent Diffusion Model (LDM), ‘Magic Clothing’, which is tailored for the task of clothing-driven image synthesis. This work introduces a garment extractor to capture detailed garment features and incorporate them into a pre-trained LDM using self-attention fusion to preserve garment details during the synthesis process. For evaluating the consistency between the target image and the source garment, this work introduces a robust metric called Matching Point-LPIPS (MP-LPIPS). Experiments demonstrate that Magic Clothing can achieve good performance under various conditions of control, displaying its effectiveness in garment-driven image synthesis.

**Strengths:**

1. this work is well-written and logical, and the author's work can be easily understood.
2. the author clearly describes the methodology designed in the paper, giving details of the experimental set-up.
3. this work performs a large number of qualitative experiments that demonstrate the excellent qualitative performance of the method. In addition, the authors have placed source code in the supplementary material, which increases the reproducibility of this work.

**Limitations:**

1. the template used in the paper does not quite match this year's conference and there are watermarks in the paper which the authors should check carefully.
2. this work is not more improved from the common Diffusion+Clip-Prompt approach to VTON and the novelty of this work is lacking.
3. there are only two tables of quantitative experiments in the main text, lack of more comparative experiments and ablation analysis of key components. The quantitative analysis of this work is insufficient.
4. the description and details of the Joint Classifier-free Guidance are lacking in the paper, and the section is lacking in the contribution and innovation of this work.
5. This work lacks experimental comparisons with some recently developed open source Diffusion-based methods, such as IDM-VTON, StableVITON, OOTDiffusion, and other related work.

**Suitability:**

2

---

### Meta-Review · Area_Chair_yj64 · 2024-07-05

**Recommendation:** Accept (Poster)
**Confidence:** 5

**Metareview:**

This paper introduces a latent diffusion model-based framework for virtual try-on, which can process diverse generation conditions such as texts, poses, and subject models. A garment extractor is designed to extract garment features, which are fused with features of other conditions for generations via attention modules and joint classifier-free guidance. The proposed framework is validated on the VITON-HD dataset.

Overall, the proposed framework is interesting and straightforward. The pre-rebuttal ratings of this paper were mixed, including two borderline rejects and one weak accept. The authors did an excellent job during the rebuttal and addressed most of the concerns of the reviewers. The final ratings of this paper are all positive (two borderline accepts and one weak accept). The AC confirmed that this paper can be revised and meet the acceptance bar of the conference. Hence, the final recommendation of this paper is accepted (poster).